

# Earthworms neutralize the influence of components of particulate pollutants on soil extracellular enzymatic functions in subtropical forests

Junbo Yang[1,2], Jingzhong Lu[2], Yinghui Yang[1], Kai Tian[3], Xiangshi Kong[4], Xingjun Tian[1,5] and Stefan Scheu[2,6]

[1] State Key Laboratory of Pharmaceutical Biotechnology, School of Life Sciences, Nanjing University, Nanjing, Jiangsu Province, China
[2] Johann Friedrich Blumenbach Institute of Zoology and Anthropology, University of Göttingen, Göttingen, Lower Saxony, Germany
[3] College of Life Science and Agricultural Engineering, Nanyang Normal University, Nanyang, Henan Province, China
[4] Key Laboratory for Ecotourism of Hunan Province, School of Tourism and Management Engineering, Jishou University, Jishou, Hunan Province, China
[5] College of Eco-Environmental Engineering, Qinghai University, Xining, Qinghai Province, China
[6] Centre of Biodiversity and Sustainable Land Use, University of Göttingen, Göttingen, Lower Saxony, Germany

Corresponding authors
Xingjun Tian, tianxj@nju.edu.cn
Stefan Scheu, sscheu@gwdg.de

## ABSTRACT

Human activities are increasing the input of atmospheric particulate pollutants to forests. The components of particulate pollutants include inorganic anions, base cations and hydrocarbons. Continuous input of particulate pollutants may affect soil functioning in forests, but their effects may be modified by soil fauna. However, studies investigating how soil fauna affects the effects of particulate pollutants on soil functioning are lacking. Here, we investigated how earthworms and the particulate components interact in affecting soil enzymatic functions in a deciduous (*Quercus variabilis*) and a coniferous (*Pinus massoniana*) forest in southeast China. We manipulated the addition of nitrogen (N, ammonium nitrate), sodium (Na, sodium chloride) and polycyclic aromatic hydrocarbons (PAHs, five mixed PAHs) in field mesocosms with and without *Eisenia fetida*, an earthworm species colonizing forests in eastern China. After one year, N and Na addition increased, whereas PAHs decreased soil enzymatic functions, based on average Z scores of extracellular enzyme activities. Earthworms generally stabilized soil enzymatic functions via neutralizing the effects of N, Na and PAHs addition in the deciduous but not in the coniferous forest. Specifically, earthworms neutralized the effects of N and Na addition on soil pH and the effects of the addition of PAHs on soil microbial biomass. Further, both particulate components and earthworms changed the correlations among soil enzymatic and other ecosystem functions in the deciduous forest, but the effects depended on the type of particulate components. Generally, the effects of particulate components and earthworms on soil enzymatic functions were weaker in the coniferous than the deciduous forest. Overall, the results indicate that earthworms stabilize soil enzymatic functions in the deciduous but not the coniferous forest irrespective of the type of particulate components. This suggests that earthworms may neutralize the influence of atmospheric particulate pollutants on ecosystem functions, but the neutralization may be restricted to deciduous forests.

# INTRODUCTION

The secretion of extracellular enzymes is an important means by which microorganisms regulate nutrient cycling in soil (*Luo, Meng & Gu, 2017*), but their secretion is affected by atmospheric pollutants caused by human activities in particular in urban regions (*Bardgett & Wardle, 2010*; *Lin et al., 2017*). In addition, the secretion and activity of extracellular enzymes is also affected by soil fauna such as earthworms (*Hoang et al., 2016*). The European earthworm species *Eisenia fetida*, common in compost heaps, is increasingly colonizing forests of urban regions in eastern China (*Huang, Zhang & Gao, 2003*; *Aira, Monroy & Domínguez, 2006*). However, the effect of *E. fetida* on soil extracellular enzyme activity in urban forests especially under the addition of atmospheric pollutants remains elusive (*Walker, 1992*; *Bardgett & Wardle, 2010*; *Blankinship, Niklaus & Hungate, 2011*). This limits our understanding of the role of these earthworms in urban forests under increasing atmospheric pollutants and hampers the development of bioremediation strategies, *e.g.*, the application of earthworms to restore terrestrial ecosystems contaminated by atmospheric pollutants.

Soil extracellular enzymes are key to the functioning of terrestrial ecosystems as they drive nutrient cycling (*Sinsabaugh, 2010*; *Burns et al., 2013*). For example, litter decomposition needs the catalysis of enzymes such as polyphenol oxidase and cellobiohydrolase, two enzymes driving the decomposition of lignin and cellulose, respectively (*Naseby, Pascual & Lynch, 2000*; *Sinsabaugh, Carreiro & Repert, 2002*). Soil extracellular enzymes can be divided into four types, *i.e.,* enzymes involved in carbon (C), nitrogen (N) and phosphorus (P) cycling as well as enzymes such as oxidases (O) involved in the breakdown of complex molecules (*Kizilkaya et al., 2011*; *Xiao et al., 2018*). Since extracellular enzymes are driving different processes in soil they may serve as indicators for the effect of atmospheric pollutants and soil fauna on ecosystem functions (*Maestre et al., 2012*; *Xiao et al., 2018*; *Liu et al., 2019*).

Industrial emission and vehicular exhaust transported by air result in the deposition of a variety of particulate pollutants into terrestrial ecosystems; the components of these pollutants include inorganic anions, base cations and hydrocarbons (*Chan & Yao, 2008*), for example, $NO_3^-$, $Na^+$ and PAHs (*Ye et al., 2003*). Another important source of particulate pollutants is fugitive dust, *Hao et al. (2005)* reported that fugitive dust and industrial dust contributes about 49% and 28% to particulate matters of <10 μm diameter (PM10) in Beijing. The use of fertilizers in arable systems and salt on roads increased the $NO_3^-$ and $Na^+$ contents of soils (*Tiwari & Rachlin, 2018*), adding to the deposition of fugitive dust. Particulate pollutants are transported by air into the surroundings of cities (*Chan & Yao, 2008*) and enter soils by precipitation (*Anderson & Downing, 2006*). In addition, the canopy of forests may collect considerable amounts of pollutants because of their rough surfaces, which later, *via* shedding of leaves, enter soils where they may accumulate
(*Belis, Offenthaler & Weiss, 2011*). The input and accumulation of these pollutants is likely to affect soil functioning, but their effects depend on deposition rates. At low rates, the components of particulate pollutants may increase soil enzyme functions (EFs), *e.g.,* Na shortage is limiting microbial activity in soil of inland forests and the addition of Na has been shown to increase soil EFs at low concentrations. However, at high rates each of N, Na and PAHs detrimentally affect soil EFs, although their effects may vary with the type of soil EFs (*Muckian et al., 2007*; *Lin et al., 2017*; *Xiao et al., 2018*; *Ji et al., 2020*). Despite being well-studied separately, the effects of N, Na and PAHs addition on soil EFs have not been investigated in concert with earthworms. This, however, is important for understanding the role of earthworms on soil functions with increasing input of the particulate pollutants.

Earthworms affect belowground processes in multiple ways including digging burrows, fragmenting litter and secreting mucus (*Marhan & Scheu, 2006*; *Szlavecz et al., 2011*; *Hoang et al., 2016*). *Eisenia fetida* is well known from compost heaps where it strongly affects enzyme activity resulting in the acceleration of the composting process (*Aira, Monroy & Domínguez, 2006*). This species is very active and reproduces rapidly resulting in fast population growth (*Gunya & Masika, 2022*). However, *E. fetida* also lives in natural habitats in southern Europe and increasingly also in eastern China, in particular close to urban areas, where it colonizes forest floors as epigeic species (*Pižl, 2002*; *Huang, Zhang & Gao, 2003*; *Lu & Lu, 2015*; *Koubová et al., 2015*; *Geraskina, 2016*). Epigeic earthworms predominantly live in the litter layer where they regulate biochemical processes (*McLean & Parkinson, 1997*; *Asshoff, Scheu & Eisenhauer, 2010*). Besides, *E. fetida* is able to live in highly contaminated soil and may even contribute to the decontamination of soils (*Rodriguez-Campos et al., 2014*). However, studies investigating effects of *E. fetida* on soil EFs in urban forests are lacking, but such studies are important for understanding the consequences of the colonization urban forests by *E. fetida*.

Previous studies focused on the toxicological effects of environmental pollutants on earthworms (*Nam et al., 2015*), and the bioaccumulation and bioremediation by earthworms (*Rodriguez-Campos et al., 2014*). Only few laboratory studies investigated how pollutants affect soil enzyme activities in the presence of earthworms (*Yin et al., 2003*; *Li et al., 2022*). However, the results of these laboratory experiments may not fully represent the actual effects of earthworms in forests considering the multiple pathways by which soil processes are affected by earthworms and their dependency on environmental factors. Field mesocosm experiment may be an appropriate way to reveal the actual effects of earthworms on ecosystem functions in forests (*Shao et al., 2017*).

In forests, earthworms may affect soil pH and microbial activity *via* gut and cast-associated processes (*Devliegher & Verstraete, 1995*; *Drake & Horn, 2007*). In particular, earthworms may increase the rate of soil nutrient cycling and microbial biomass by accelerating litter decomposition, and forming biopores and microbial hotspot areas (*Hoang et al., 2016*; *Yang et al., 2023*). Since particulate pollutants detrimentally affect microbial activity by promoting soil acidification and leaching of nutrients (*Lu et al., 2014*; *Yang et al., 2023*), earthworms may attenuate their detrimental influence by increasing soil pH and improving nutrients availability. As the abundance of earthworms varies among different types of forests, it may be necessary to investigate their effects at varying density
(*Cortez, 1998*; *Szlavecz et al., 2011*). Further, the effects of N, Na and PAHs on soil functions may vary with forest types (*Lin et al., 2017*; *Ji et al., 2020*; *Yang et al., 2023*). Therefore, to understand interactions between earthworms and pollutants in different forests at close to natural settings, experimental studies need to mimic the density of earthworms in the respective forests.

For studying interactions between earthworms and particulate pollutants at high input rates into different forests, we performed a field experiment in a deciduous (*Quercus variabilis*) and a coniferous (*Pinus massoniana*) forest with and without the addition of the earthworm species *E. fetida* in southeast China. We hypothesized that (1) the addition of N, Na and PAHs detrimentally affect soil EFs and (2) earthworms promote soil EFs and counteract negative effects of N, Na and PAHs addition on soil EFs.

## MATERIALS & METHODS

### Study sites

We performed a field experiment in deciduous and coniferous subtropical forests in Zijin Mountain close to the city of Nanjing, Southeast China (32°4′N, 118°51′E), from April 2018 to May 2019 (day 0 to 365). *Quercus variabilis* and *Pinus massoniana* dominated in the deciduous and coniferous forest, respectively, and *Parthenocissus quinquefolia* and *Carex* spp were the dominant ground floor species (*Tian et al., 2018*). The mean annual air temperature and rainfall were 15.4 °C and 1106 mm, respectively. Soil pH ranged from 5.3 to 5.6. The bedrock was sandstone and shale covered by a humus layer rich in organic matter and nutrients, soils are humic cambisols. The deciduous and coniferous forests were located at similar altitudes (65 and 175 m, respectively) and about 900 m away from each other. For more details on the study sites see Table 1.

### Experimental design

In each forest, we identified a core area of 30 m × 20 m for establishing the experiment. We investigated the effects of particulate components (control, N, Na, PAHs) and earthworms (with and without) on soil EFs. In each forest, each treatment was replicated four times resulting in 32 experimental units (4 particulate components treatments including the control × 2 earthworm treatments × 4 replicates) comprising individual mesocosms. For each unit, we first dug up a pit of an area of 0.5 m × 0.5 m to a depth of 0.2 m. Earthworms, herb seedlings and fine roots were hand-sorted from the excavated litter and soil. Thus, the soil EFs investigated were mainly contributed by soil microorganisms (not roots). Then, we placed a square nylon bag (1 m × 1 m, 0.16 mm mesh size) into the pit and filled back the excavated soil to fit the outside layers. Finally, we used a zipper at the top of the bag to close it and covered it with leaf litter. The mesocosms were spaced at least 3 m from each other and set away from thick roots (for details see Fig. S1). As dry soil is more convenient for picking earthworms, the installment of the mesocosms was conducted on sunny days from February to April 2018.

For each of the particulate component treatments we at least doubled the input compared to the control. Based on earlier studies (*Qasemian et al., 2012*; *Jia et al., 2015*; *Lin et al., 2017*), the amount of particulate components added was assumed to affect soil enzyme

**Table 1** **Site conditions and litter traits of the deciduous and coniferous forests.** Means ± SD. Different letters indicate significant differences; $t$-test ($p < 0.05, n = 5$). Values of soil C, soil N and soil C/N ratio were taken from *Tian et al. (2018)*; values of concentrations of PAHs in soil were taken from *Wang et al. (2015)*.

| | Deciduous | Coniferous |
|---|---|---|
| **Site conditions** | | |
| Latitude (°N) | 32.0555 | 32.0635 |
| Longitude (°E) | 118.8736 | 118.8727 |
| Elevation (m a.s.l.) | 65 | 175 |
| Slope (°) | 5 | 15 |
| Soil pH | 5.30 ± 0.19a | 5.62 ± 0.48a |
| Soil moisture (%) | 24.25 ± 1.86a | 21.48 ± 0.40b |
| Soil C (g kg$^{-1}$) | 20.30a | 15.40b |
| Soil N (g kg$^{-1}$) | 1.30a | 1.10a |
| Soil C: N | 15.52a | 14.39a |
| Soil organic matter (g kg$^{-1}$) | 35.00a | 26.55b |
| **PAH concentrations** ($\mu$g kg$^{-1}$) | | |
| Fluoranthene (Flu) | 519 ± 726 | |
| Pyrene (Pyr) | 429 ± 596 | |
| Chrysene (Chr) | 328 ± 395 | |
| Benzo[a]pyrene (BaP) | 278 ± 379 | |
| Phenanthrene (Phe) | 259 ± 314 | |
| $\sum$16PAHs | 3,330 ± 4250 | |
| **Litter traits** | *Quercus variabilis* | *Pinus massoniana* |
| Lignin (%) | 31.20 ± 1.08b | 40.60 ± 0.77a |
| Total C (%) | 49.50 ± 0.83b | 51.30 ± 1.26a |
| Total N (%) | 1.27 ± 0.07a | 0.85 ± 0.08b |
| Lignin: N | 33.31 ± 1.77b | 60.53 ± 5.77a |
| C: N | 24.63 ± 2.68b | 48.47 ± 6.86a |

activities. The amount of N added was equivalent to 47 kg N ha$^{-1}$ y$^{-1}$, which resembles the average amount of N deposited in the urban region of Nanjing (*Lin et al., 2017*), thereby doubling the nitrogen deposition compared to the control. The amount of Na added was equivalent to 39 g Na m$^{-2}$ y$^{-1}$, which is much higher than the natural input, but in the range occurring close to roadsides in the region of Nanjing (*Jia et al., 2015*). A higher level of Na was added to account for extra input of Na by human activities such as road salt (*Li et al., 2016*; *Tiwari & Rachlin, 2018*). In the PAHs treatment, a mixture of five PAHs including fluoranthene, pyrene, chrysene, benzo[a]pyrene and phenanthrene was added equivalent to a total amount of 0.51 g m$^{-2}$ y$^{-1}$, with individual amounts of 146, 121, 92, 78 and 73 mg m$^{-2}$ y$^{-1}$, respectively. The five PAHs used account for 54% of the mass of 16 prioritized PAHs in urban soils of Nanjing and Zijin Mountain, and the added amount of PAHs resembled the amount deposited to soils in the urban region of Nanjing within one year (Table 1, *Wang et al., 2015*), again, doubling the input of the five PAHs to the soil of the PAHs treatments.

N, Na and PAHs treatments received 500 mL aqueous solutions/suspensions of $NH_4NO_3$, NaCl and PAHs every 35 days, the control treatment received 500 mL distilled water. Solutions and distilled water were sprayed onto the litterbags (for details see below and Fig. S1) and soil surface of the mesocosms prior to closing the zipper of the nylon bags by using bottles with spray head (Amway 00483, Ada, MI, USA). The solution of PAHs was prepared by successively dissolving benzo[a]pyrene, phenanthrene, fluoranthene and pyrene in 2.5 mL dimethyl sulfoxide; chrysene was dissolved in 2.5 mL ethanol, then the ethanol was poured into dimethyl sulfoxide and 495 mL distilled water was added. As the PAHs did not fully solve, the spray bottles were shaken while spraying to ensure that they were fully added.

To investigate effects of earthworms at densities close to their abundance in the different forests, each mesocosm of the earthworm treatments in the deciduous forest received 60 adult individuals of *E. fetida*, whereas earthworm treatments in the coniferous forest received 20 individuals per mesocosm. The numbers added resembled the abundance of earthworms in the two forests as investigated by hand sorting in 2018 (Table 2). The body length of individuals added was 80–100 mm. According to the investigations in 2018, from the litter layer to 20 cm soil depth, *E. fetida* accounted for $52.5 \pm 15.1\%$ and $64.4 \pm 19.0\%$ of total earthworm abundance, with their biomass averaging $4.7 \pm 1.1$ and $2.3 \pm 1.1$ g m$^2$ in the deciduous and coniferous forests, respectively (mean $\pm$ SD, $n = 5$; Table 2). We placed 10 litterbags filled with litter of *Q. variabilis* and *P. massoniana* (5 coarse and 5 fine mesh litterbags with 5.0 mm and 0.2 mm mesh size, respectively, 8 g dry litter for each coarse bag and 4 g litter for each fine bag) on the surface of the soil inside each mesocosm in both the deciduous and coniferous forest. Coarse and fine mesh litterbags were used to investigate the contribution of soil fauna and microorganisms to litter decomposition, respectively (for details see Fig. S1). To validate the earthworm treatments, we dug up the soil of mesocosms, picked earthworms from the mesocosms by hand, counted them and then placed them back during the period from May to August 2019. Soil and litter samples were taken at day 70, 140, 210, 280 and 365 resulting in a total of 640 soil (0–5 cm depth) and litter samples.

## Soil properties and litter mass loss

Soil cores underneath litterbags were taken to measure soil properties at 0–5.0 cm soil depth. Fresh soil samples were sieved through 1.0 mm mesh and then stored at 4 °C. Soil moisture was determined gravimetrically. Soil pH was measured in a 1 : 2.5 soil to water solution by using a pH meter (Mettler Toledo, Columbus, OH, USA). Soil microbial biomass was determined by substrate-induced respiration (SIR) following *Bailey et al. (2002)*, for details see *Lin et al. (2017)*. Nine soil enzyme activities were measured including three C related enzymes ($\beta$-1,4-glucosidase, E.C. 3.2.1.21; $\beta$-1,4-xylosidase, E.C. 3.2.1.37; cellobiohydrolase, E.C. 3.2.1.91), two N related enzymes (nitrate reductase, E.C. 1.7.99.4; urease, E.C. 3.5.1.5), two P related enzymes (acid phosphatase, E.C. 3.1.3.2; alkaline phosphatase, E.C. 3.1.3.1) and two enzymes processing complex C compounds such as lignocellulose (O enzymes; peroxidase, E.C. 1.11.1.7; polyphenol oxidase, E.C. 1.10.3.2) (*Saiya-Cork, Sinsabaugh & Zak, 2002*; *Lin et al., 2017*). Activities of C and O

**Table 2  Abundance and biomass of earthworms in mesocosms of the deciduous and coniferous forest with and without addition of *Eisenia fetida*.** Means ± SD, $n = 4$ ($n = 5$ for initial values). Initial sampling was in February 2018, the later samplings were from May to August 2019.

| Treatments | Deciduous - Earthworms | Deciduous + Earthworms | Coniferous - Earthworms | Coniferous + Earthworms |
|---|---|---|---|---|
| **Total abundance (number)** | | | | |
| Initial | 0 | 60.60 ± 9.71 | 0 | 19.60 ± 8.08 |
| Control | 3.67 ± 6.35 | 10.00 ± 4.69 | 0.50 ± 0.58 | 3.75 ± 3.20 |
| N addition | 3.75 ± 4.86 | 15.75 ± 8.73 | 1.75 ± 2.36 | 4.00 ± 0.82 |
| Na addition | 1.25 ± 1.26 | 8.00 ± 4.24 | 4.00 ± 4.62 | 9.50 ± 5.00 |
| PAHs addition | 2.00 ± 2.16 | 14.75 ± 14.08 | 1.25 ± 1.50 | 4.75 ± 1.26 |
| **Total biomass (dry mass, g)** | | | | |
| Initial | 0 | 1.69 ± 0.41 | 0 | 0.81 ± 0.25 |
| Control | 0.04 ± 0.06 | 0.50 ± 0.37 | 0.01 ± 0.01 | 0.08 ± 0.05 |
| N addition | 0.07 ± 0.05 | 0.95 ± 1.02 | 0.05 ± 0.06 | 0.11 ± 0.03 |
| Na addition | 0.03 ± 0.03 | 0.30 ± 0.08 | 0.06 ± 0.07 | 0.41 ± 0.28 |
| PAHs addition | 0.04 ± 0.06 | 0.97 ± 1.13 | 0.04 ± 0.07 | 0.12 ± 0.07 |
| ***Eisenia* abundance (number)** | | | | |
| Initial | 0 | 31.00 ± 7.18 | 0 | 12.20 ± 5.07 |
| Control | 0 | 7.25 ± 2.75 | 0 | 3.50 ± 2.89 |
| N addition | 1.50 ± 1.73 | 7.00 ± 10.80 | 1.75 ± 2.36 | 3.50 ± 1.29 |
| Na addition | 0.25 ± 0.50 | 4.25 ± 3.40 | 2.00 ± 2.45 | 5.75 ± 2.63 |
| PAHs addition | 0.25 ± 0.50 | 7.25 ± 9.98 | 1.00 ± 1.41 | 4.00 ± 1.41 |
| ***Eisenia* biomass (dry mass, g)** | | | | |
| Initial | 0 | 1.18 ± 0.28 | 0 | 0.59 ± 0.27 |
| Control | 0 | 0.36 ± 0.17 | 0 | 0.08 ± 0.05 |
| N addition | 0.05 ± 0.05 | 0.52 ± 0.95 | 0.05 ± 0.06 | 0.09 ± 0.05 |
| Na addition | 0.01 ± 0.02 | 0.22 ± 0.13 | 0.04 ± 0.05 | 0.14 ± 0.07 |
| PAHs addition | 0.01 ± 0.01 | 0.72 ± 0.99 | 0.04 ± 0.07 | 0.11 ± 0.07 |

related enzymes were determined by using a microplate spectrophotometer (Tecan Safire2, Männedorf, Switzerland), the other enzyme activities were determined by using a UV spectrophotometer (JingHua, Shanghai, China). For measuring C related enzymes, nitrate reductase, urease, P related enzymes and O related enzymes, 4-nitrophenyl-b-D-linked (PNPX) substrates, $KNO_3$, urea, disodium phenyl phosphate and l-3,4-dihydroxy-phenylalanine were used as substrates, respectively. Details on the measurements are given in supplemental material.

Since the effects of particulate components and earthworms on soil EFs were investigated at different times of litter decomposition, the rates of litter mass loss were calculated. The litter materials from the litterbags were cleaned using distilled water and then dried at 60 °C for 72 h. Total C and N of the litter was measured at days 70, 210 and 365 using an elemental analyzer (Elemental Vario Micro, Langenselbold, Germany). Based on these data, we calculated litter mass, C and N loss and expressed them as percentages of initial.

## Statistical analyses

All analyses were performed using R v4.0.5 (https://www.r-project.org/). Data of the deciduous and coniferous forest were analyzed separately. We used generalized linear models with Poisson distribution to analyze the abundance and biomass of earthworms at the end of the experiment. When the models were over- or under-dispersed, quasi-Poisson distribution was used.

We used average $Z$ scores of enzyme activities to indicate soil EFs (*Maestre et al., 2012*). Five types of soil EFs were calculated, total EF, EF-C, EF-N, EF-P and EF-O referring to the average $Z$ scores of all nine enzyme activities, three C enzyme activities, two N enzyme activities, two P enzyme activities and two O enzyme activities, respectively. We then used permutational multivariate analysis of variance (PERMANOVA) to analyze soil enzyme activities and linear mixed effects models (LMMs) to analyze each of the five soil EFs, soil moisture, pH and microbial biomass. In PERMANOVA and LMM, particulate components (control, N, Na, PAHs), earthworms (with and without), mesh size (fine and coarse) and time (five sampling dates) were treated as fixed factors. In LMM, mesh size was nested in mesocosms and included as random factor to account for non-independence of litterbags within mesocosms and repeated sampling. We used planned contrasts to evaluate the effect sizes of particulate components, earthworms and mesh size with the control as reference. The difference of estimated marginal means of soil EFs resembles Cohen's d as the standard deviation of the $z$ score is 1. Contrasts of soil moisture, pH and microbial biomass are analogous to log response ratios as the response variables were $\log(x+1)$ transformed to improve normality (*Piovia-Scott et al., 2019*). We used 'nlme' to fit mixed-effects models and 'emmeans' for planned contrasts.

We used one-way ANOVA and unpaired $t$-test to analyze the difference in enzyme activities between particulate components and earthworm treatments, respectively. Non-parametric tests, *i.e.*, Kruskal-Wallis test and Mann-Whitney test, were used if the data did not fit normality. Percentages of enzyme activity were calculated as $(|t-c|/c) \times 100\%$, with $t$ and $c$ the enzyme activities of the treatment and control, respectively. Soil EFs are closely related to other ecosystem processes, *e.g.*, litter decomposition (*Sinsabaugh, 2010*; *Burns et al., 2013*). We inspected changes in the fit of correlations among soil EFs, litter decomposition and other soil properties using Pearson correlations with and without earthworms to understand multiple pathways linking earthworms with ecosystem functions in each pollutant treatment (*Muckian et al., 2007*; *Liu et al., 2019*; *Ji et al., 2020*).

# RESULTS

## Earthworm abundance

The abundance and biomass of total earthworms as well as the abundance and biomass of *E. fetida* was significantly increased in the treatments with addition of *E. fetida* in both the deciduous and coniferous forest ($P < 0.001$, Chi$^2$ >14.14, $df = 1$; Table S1). On average the abundance of total earthworms was increased by factors of 2.7 to 7.4 and by factors of 2.3 to 7.5 in the deciduous and coniferous forest, respectively (Table 2). With *E. fetida* addition, the percentage of dry mass of *E. fetida* to total earthworm dry mass ranged from

**Table 3  Results of linear mixed-effects models on the effects of particulate components, earthworms, mesh size and time on *Z* scores of total soil enzymatic activities in deciduous and coniferous forests.** For the *F*- and *P*-values of linear mixed-effects models on the effects of the factors studied on *Z* scores of individual soil enzymatic activities of C, N, P and O in the deciduous and coniferous forests see Tables S3 and S4, respectively; df, numerator and denominator degrees of freedom. Significant effects ($P < 0.05$) are given in bold.

| | Deciduous | | | Coniferous | | |
|---|---|---|---|---|---|---|
| | df | F | P | df | F | P |
| Particulate components (PC) | 3,23 | 4.35 | **0.014** | 3,24 | 1.85 | 0.166 |
| Earthworms (E) | 1,23 | 1.67 | 0.209 | 1,24 | 0.58 | 0.454 |
| Mesh size (M) | 1,23 | 11.56 | **0.002** | 1,24 | 16.96 | **<0.001** |
| Time (T) | 4,184 | 30.28 | **<0.001** | 4,192 | 323.43 | **<0.001** |
| PC × E | 3,23 | 5.08 | **0.008** | 3,24 | 0.29 | 0.835 |
| PC × M | 3,23 | 1.88 | 0.161 | 3,24 | 0.26 | 0.854 |
| E × M | 1,23 | 0.02 | 0.884 | 1,24 | 0.72 | 0.404 |
| PC × T | 12,184 | 7.94 | **<0.001** | 12,192 | 4.48 | **<0.001** |
| E × T | 4,184 | 3.79 | **0.005** | 4,192 | 0.98 | 0.421 |
| M × T | 4,184 | 25.18 | **<0.001** | 4,192 | 16.09 | **<0.001** |
| PC × E × M | 3,23 | 6.11 | **0.003** | 3,24 | 0.70 | 0.559 |
| PC × E × T | 12,184 | 2.38 | **0.007** | 12,192 | 1.67 | 0.077 |
| PC × M × T | 12,184 | 2.30 | **0.009** | 12,192 | 3.91 | **<0.001** |
| E × M × T | 4,184 | 2.41 | 0.051 | 4,192 | 0.37 | 0.829 |
| PC × E × M × T | 12,184 | 1.36 | 0.190 | 12,192 | 0.86 | 0.589 |

54.7% to 99.2% except in the Na addition treatment in the coniferous forest (34.1%). With *E. fetida* addition, the survival rates of *E. fetida* in control, N, Na and PAHs treatments were 12.1%, 11.7%, 7.1% and 12.1% in the deciduous forest and 17.5%, 17.5%, 28.8% and 20.0% in the coniferous forest, respectively (Table 2).

## Soil enzymatic functions

In the deciduous forest, particulate components significantly affected soil total EFs ($F = 4.35$, $P < 0.05$, Table 3). Overall, the addition of N and Na increased soil total EF, EF-C, and EF-N (Table S3, Fig. 1). In the treatments without addition of earthworms, particulate components significantly increased soil total EF, EF-C and EF-N after 280 and 365 days, with the increase being highest in the Na and lowest in the PAHs treatments (Table 3, Figs. 2 and 3). However, at day 70, Na significantly decreased EF-P and EF-O, PAHs significantly decreased soil total EF and soil EF-C. In total, Na increased activities of cellobiohydrolase, $\beta$-1,4-xylosidase, nitrate reductase and urease by 27.8%, 32.8%, 57.1% and 63.0%, but decreased activities of alkaline phosphatase and peroxidase by 28.0% and 40.0%, respectively (Fig. 4, Fig. S7). N increased activities of $\beta$-1,4-xylosidase and urease by 25.0% and 24.6%, whereas PAHs decreased activities of cellobiohydrolase, $\beta$-1,4-glucosidase and $\beta$-1,4-xylosidase by 11.1%, 10.2% and 9.4%, respectively.

Factor analysis also showed that particulate components significantly affected soil total EFs, in the deciduous but not the coniferous forest, although N and Na significantly increased EF-N after 280 and 365 days in the coniferous forest (Table 3, S2, Figs. 2 and 3).

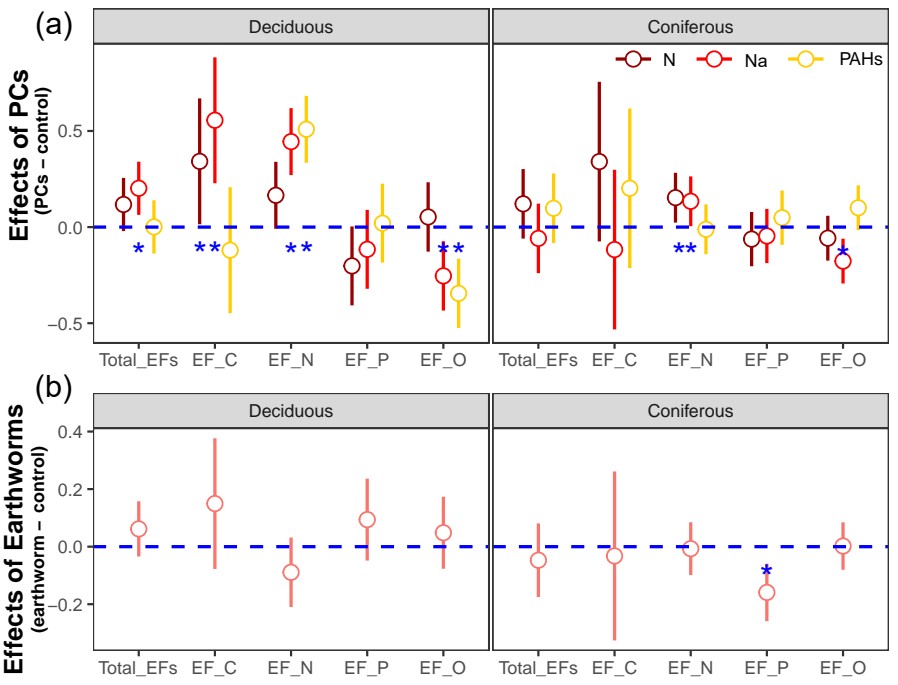

**Figure 1** **Effects of particulate components (A) and earthworms (B) on soil enzyme activities in decid-uous and coniferous forests as indicated by estimated $Z$ scores (see Methods).** Means with 95% confi-dence intervals; in (A) effect sizes were averaged across mesh size (coarse and fine), earthworm treatment (with and without) and sampling dates (70, 140, 210, 280 and 365 days), $n = 80$; in (B) effect sizes were averaged across mesh size (coarse and fine), particulate components (PCs; control, N, Na and PAHs) and sampling dates (70, 140, 210, 280 and 365 days), $n = 160$; asterisks indicate significant differences to the control ($P < 0.05$).

In total, the addition of N and Na increased urease activities in the coniferous forest by 10.0% and 13.9%, respectively (Fig. 4).

Earthworms generally did not significantly affect EFs in the coniferous forest (Table 3, Fig. S3), whereas in the deciduous forest earthworms significantly affected soil EFs, but the effect varied among particulate components for total enzyme activity, and the activity of C, N and O enzymes (significant earthworm × particulate component interaction; Table 3, Table S3, Fig. 1). In the deciduous forests, earthworms decreased the positive effects of the addition of N and Na on the EF-C and EF-N, and the negative effects of the addition of Na on EF-P and EF-O as well as the negative effects of the addition of PAHs on EF-C (Figs. 2 and 3). The reduction of the effect of particulate components by earthworms in the deciduous forest was similar in both coarse and fine mesh bags (Table 3, Fig. S2, S8, S9). Na increased the activities of $\beta$-1,4-xylosidase, alkaline phosphatase, cellobiohydrolase, nitrate reductase and urease by 2.7%, 3.4%, 3.8%, 11.9% and 30.0%, respectively, but reduced the activity of peroxidase by 7.6% (Fig. 4, Fig. S7). N increased activities of urease by 5.5%, but reduced the activity of $\beta$-1,4-xylosidase by 6.7%; PAHs increased the activities of $\beta$-1,4-xylosidase and $\beta$-1,4-glucosidase by 2.7% and 9.6%, respectively. In the coniferous

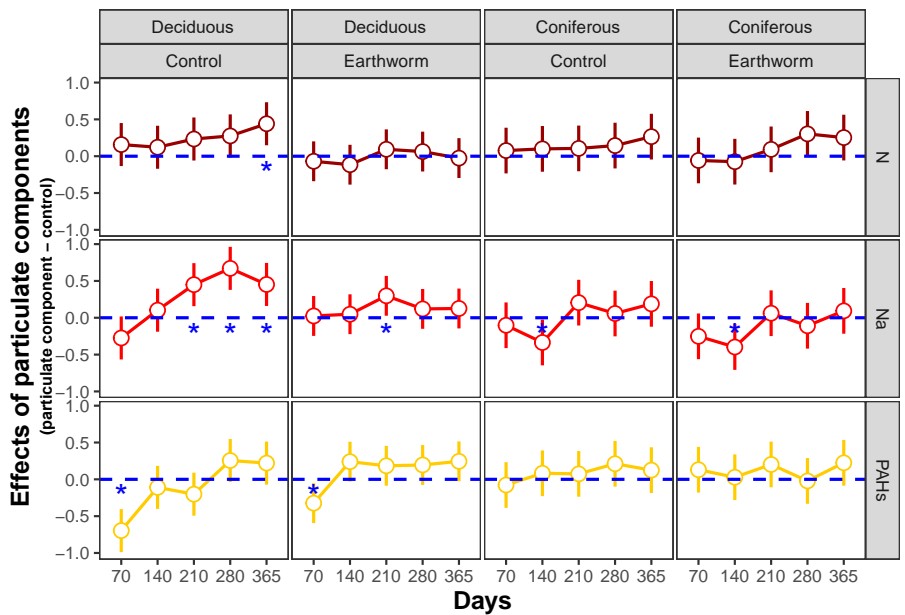

**Figure 2** Changes in estimates of Z scores of soil total enzyme activities with time as affected by different types of particulate components and earthworms in deciduous and coniferous forests. Means with 95% confidence intervals; effect sizes were averaged across mesh size (coarse and fine), $n = 8$; asterisks indicate significant differences to the control ($P < 0.05$). For the original values of Z scores of soil total enzyme activities see Fig. S2.

forest, only in the N and Na treatments earthworms increased urease activity by 7.8% and 9.2%, respectively (Fig. 4).

## Soil moisture, pH and microbial biomass

In the deciduous forest in the treatments without addition of earthworms, the addition of particulate components significantly decreased soil pH (Table S5, Fig. S10, S11). Further, the addition of Na and PAHs decreased soil microbial biomass after 70 days (Fig. S12, S13). Soil pH in the control, N, Na and PAHs treatments averaged 5.3 ±0.2, 5.0 ± 0.3, 5.0 ± 0.2 and 4.8 ± 0.2 (means ±SD across mesh size treatments and sampling times, $n = 40$), respectively (Fig. S11). After 70 days, soil microbial biomass in the control, Na and PAHs treatments averaged 42.8 ± 31.3, 30.5 ± 29.5 and 15.8 ± 15.8 (means ± SD across mesh size treatments, $n = 8$), respectively (Fig. S13).

In the coniferous forest, the addition of PAHs decreased, but Na addition increased soil pH (Fig. S10). Further, in the coniferous forest the effects of N, Na and PAHs addition on soil microbial biomass were less strong than in the deciduous forest (Fig. S12). Overall, soil pH in the control, Na and PAHs treatments averaged 4.8 ± 0.1, 5.0 ± 0.2 and 4.6 ± 0.2 (means ± SD, $n = 40$), respectively (Fig. S11).

In the treatments with earthworm addition, the negative effects of N and Na addition on soil pH, and the negative effects of Na and PAHs addition on soil microbial biomass after 70 days were less pronounced in the deciduous than in the coniferous forests (Fig. S10, S12). Overall, soil pH in the control, N, Na and PAHs treatments in the deciduous

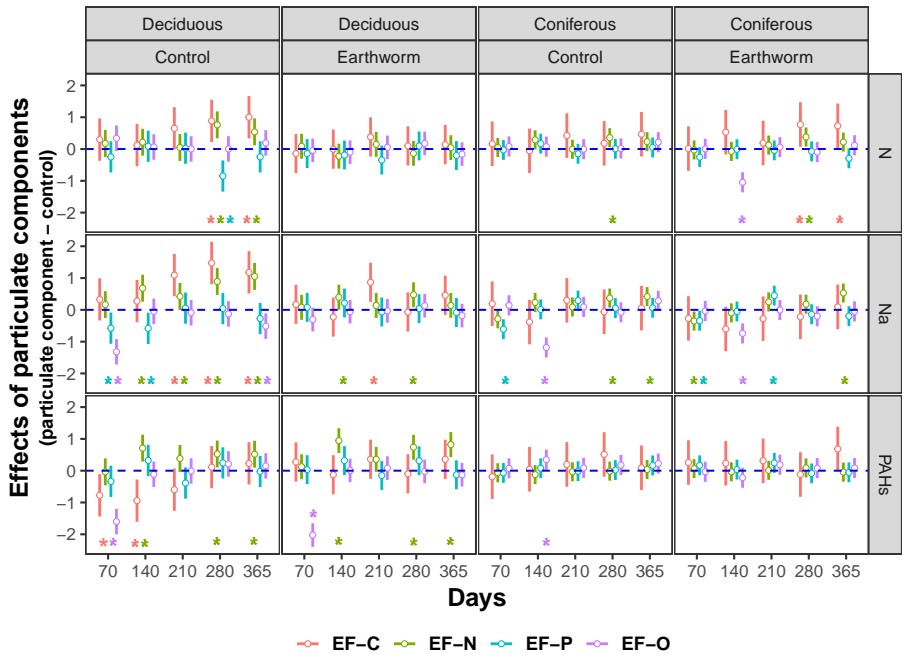

**Figure 3 Changes in estimates of *Z* scores of soil C, N, P enzymes and oxidases activities with time as affected by different types of particulate components and earthworms in deciduous and coniferous forests.** EF-C refers to three soil carbon enzyme activities, *i.e.,* $\beta$-1,4-glucosidase, $\beta$-1,4-xylosidase and cellobiohydrolase; EF-N refers to two soil nitrogen enzyme activities, *i.e.,* nitrate reductase and urease; EF-P refers to two soil phosphorus enzyme activities, *i.e.,* acid and alkaline phosphatases; EF-O refers to two soil oxidases activities, *i.e.,* peroxidase and polyphenol oxidase; means with 95% confidence intervals; effect sizes were averaged across mesh size (coarse and fine), $n = 8$; asterisks indicate significant differences to the control ($P < 0.05$). For original values of *Z* scores of soil carbon, nitrogen, phosphorus and oxidase enzyme activities see Figs. S3–S6, respectively.

forest averaged $5.2 \pm 0.2$, $5.1 \pm 0.2$, $5.2 \pm 0.2$ and $4.7 \pm 0.2$ ($n = 40$) (Fig. S11). After 70 days, soil microbial biomass in the control, Na and PAHs treatments averaged $42.8 \pm 31.3$, $30.5 \pm 29.5$ and $15.8 \pm 15.8$ (means $\pm$ SD across mesh size, $n = 8$), respectively (Fig. S13). In the coniferous forest, soil pH in the control, Na and PAHs treatments averaged $4.8 \pm 0.1$, $5.0 \pm 0.2$ and $4.6 \pm 0.1$ (means $\pm$ SD, $n = 40$), respectively (Fig. S11). In general, the addition of particulate components did not significantly affect soil moisture (Tables S5, S6, Figs. S14, 15).

## Correlation between enzymatic functions, litter decomposition and soil properties

In the deciduous forest, particulate components strengthened the correlations between soil EF-C and litter decomposition as well as soil properties, and the same was true for EF-N (Fig. 5). Earthworms decreased the effects of N and Na but increased the effect of PAHs. The addition of N, Na and PAHs strengthened the correlation between EF-N and litter mass, C and N loss; further, the addition of N strengthened the correlation between EF-C and litter mass, C and N loss. Earthworms weakened the correlations between EF-C and litter mass, C and N loss in the N addition treatment, and the correlations between EF-N

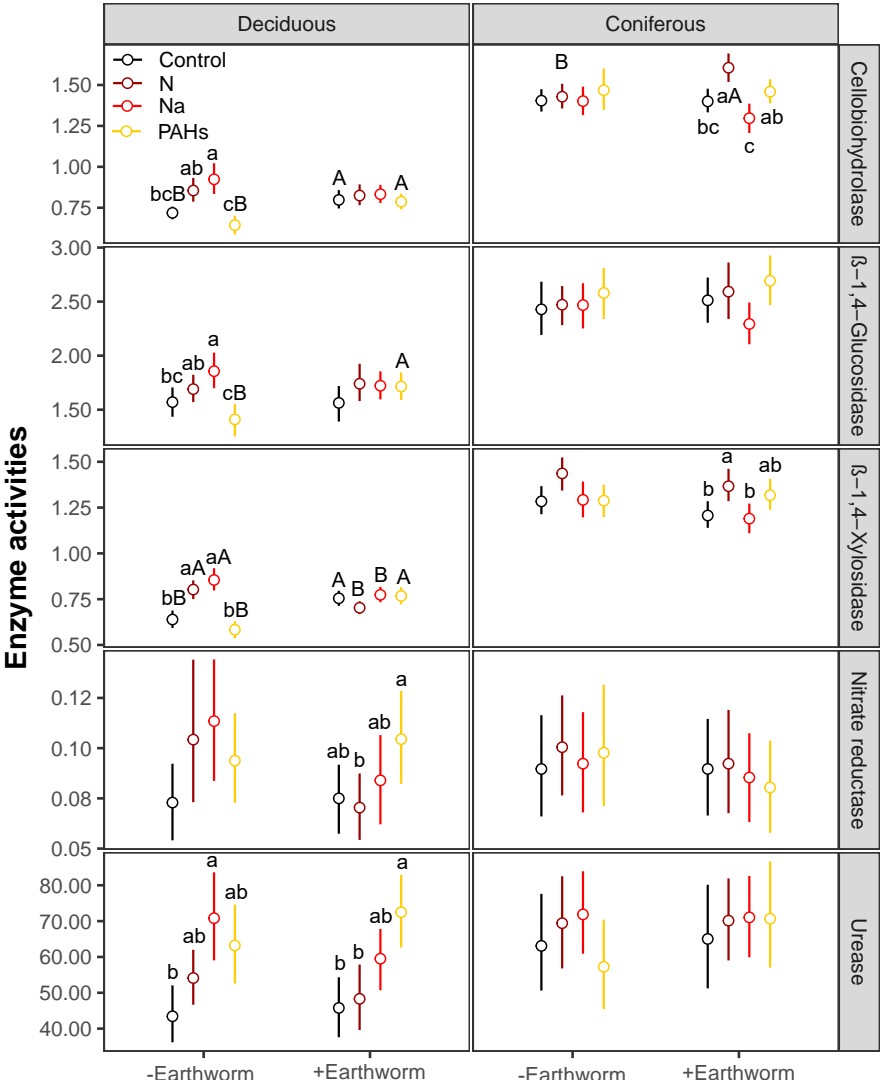

**Figure 4 Changes in carbon and nitrogen enzyme activities as affected by different types of particulate components and earthworms in the soil underneath litterbags in deciduous and coniferous forests.** Means ± SE, values were averaged across sampling dates (70, 140, 210, 280, 365 days) and mesh size (coarse and fine), $n = 40$; lower case and capital letters indicated the difference in enzyme activities in deposited compound treatments (control, N, Na and PAHs) and in earthworm treatments (without, with), respectively. For original values of soil phosphorus and oxidase enzyme activities see Fig. S7.

and litter mass loss, soil pH, moisture and microbial biomass in the Na addition treatment. Also, earthworms weakened the correlations between EF-N and litter mass loss, C loss and soil microbial biomass in the control without particulate components. Conversely, earthworms strengthened the correlations between EF-N and litter mass, C and N loss in the treatment with addition of PAHs (Fig. 5).

Contrasting the deciduous forest, the effects of particulate components on the correlations between EFs and litter decomposition as well as soil properties were less strong

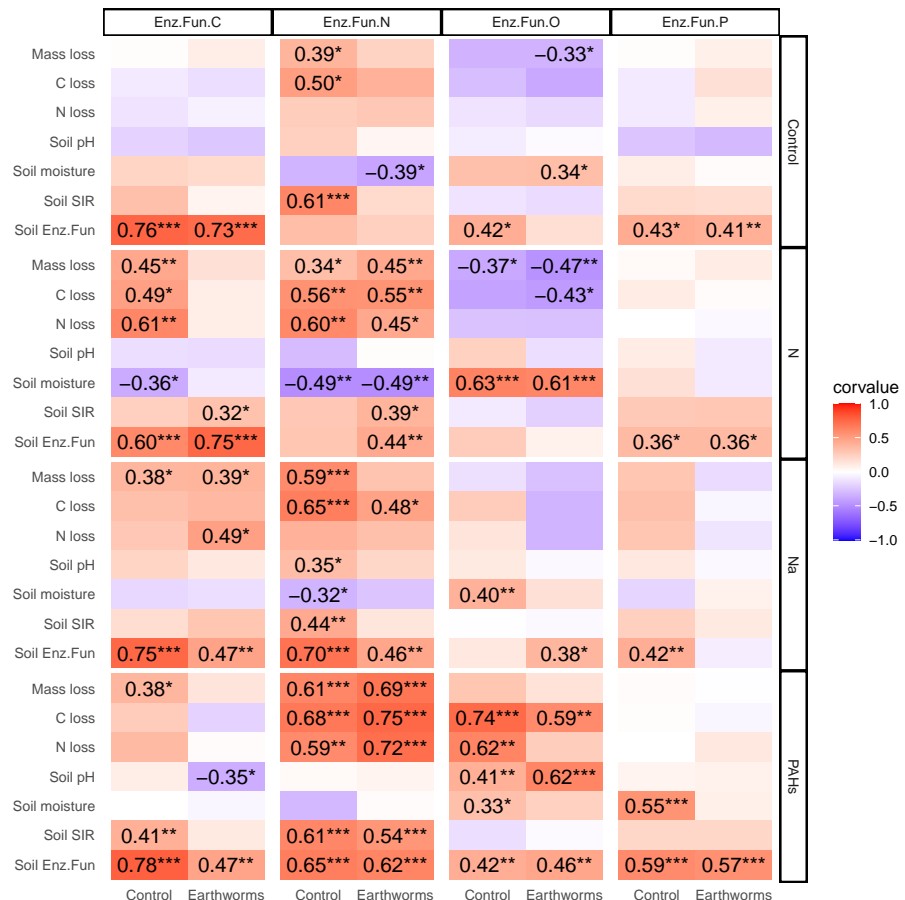

**Figure 5 Correlation matrix between different soil enzymatic functions and litter decomposition parameters as well as soil properties in particulate components and earthworm treatments in the deciduous forest.** Enz.Fun.C, Enz.Fun.N, Enz.Fun.O, Enz.Fun.P, Soil Enz.Fun refer to enzymatic functions of carbon, nitrogen, oxidase, phosphorus and total, respectively. Mass loss, C loss and N loss refer to percentage changes during decomposition of *Quercus variabilis* litter. Corvalue refers to the Pearson correlation coefficient and is given for correlations with $P < 0.05$; *, $P \leq 0.05$; **, $P \leq 0.01$; ***, $P \leq 0.001$.

in the coniferous forest (Fig. S16). Although earthworms strengthened the correlations between EF-N and soil pH as well as litter N loss in the Na and PAHs treatments, overall the modification of effects of particulate components on EFs by earthworms was less strong in the coniferous than in the deciduous forest (Fig. S16).

## DISCUSSION

Biotic interactions are key for understanding soil functioning under atmospheric particulate pollutants. We investigated the interactive effects of different particulate components and earthworms on soil EFs in forest ecosystems. Without earthworms, the effects of particulate components on soil EFs in the deciduous forest depended on the type of enzyme function. Nutrient acquisition by soil microorganisms is based on the secretion of extracellular enzymes (*Luo, Meng & Gu, 2017*) and pollutants are likely to affect microbial nutrient
acquisition by changing the secretion of enzymes. Notably, earthworms generally stabilized soil EFs and reduced the detrimental effects of particulate components on EF-C and EF-N irrespective of the type of particulate components in the deciduous forest. This may have been due to earthworms improving microbial nutrient availability thereby diminishing the detrimental effects of particulate components on the secretion of extracellular enzymes. To manipulate earthworm treatments, we mixed soil and placed it into nylon bags, which likely homogenized hotspots soil enzyme activity such as earthworm burrows and rhizosphere soil. This may have resulted in soil enzyme activity in mesocosms being lower than under natural conditions, but allowed investigating interactions between particulate components and earthworms.

## Particulate components significantly affected soil EFs

Without earthworms, PAHs decreased, whereas N and Na increased soil EFs in the deciduous forest, partly supporting our first hypothesis. Soil EFs are predicted to be sensitive to the addition of particulate components, and benefit, *e.g.*, from increased input of N according to the resource allocation theory (*Sinsabaugh & Moorhead, 1994*; *Allison & Vitousek, 2005*). Conform to these expectations, N and Na increased soil EFs in treatments without earthworms in the deciduous forest. However, the effects of particulate components on soil EFs varied among the types of particulate components studied, with PAHs generally detrimentally affecting soil EFs which is in line with earlier studies (*Klamerus-Iwan et al., 2015*; *Lipińska, Kucharski & Wyszkowska, 2019*). Effects of the addition of N, Na and PAHs on soil moisture were small in deciduous and coniferous forests, suggesting that the addition of pollutants may not change the activities of soil macrofauna and therefore also may not affect soil structure (*Brown, 1995*). By contrast, in the deciduous forest all three particulate components decreased soil pH with the effect of PAHs being strongest suggesting that the effects of these components on soil EFs were due to soil acidification. Interestingly, despite decreasing soil pH, N and Na addition increased soil EFs. Previous studies also found N and Na addition to increase soil enzyme activities (*Lin et al., 2017*; *Ji et al., 2020*). Considering the Na limitation in inland forests, Na addition may indirectly increase the energy flow through soil food webs and thereby the activity of soil enzymes (*Kaspari et al., 2009*; *Ji et al., 2020*). The fact that the addition of PAHs did not significantly increase soil EFs may have been due to their low solubility and recalcitrance (*Blakely, Neher & Spongberg, 2002*; *Jonker & vander Heijden, 2007*). It is noteworthy that the three particulate components did not uniformly decrease soil microbial biomass parallel to soil pH suggesting that their effect on EFs was due to modifying extracellular enzymes of specific microorganisms rather than detrimentally affecting the activity of the whole soil microbial community.

Effects of different types of particulate components on ecosystem functions may also vary with the structure of soil food webs (*Berg, Johansson & Meentemeyer, 2000*; *Blakely, Neher & Spongberg, 2002*; *Kaspari et al., 2014*; *Zhang et al., 2016*). Our results indicate that in absence of earthworms, the effects of N, Na and PAHs on soil EF-C and EF-N are stronger than on soil EF-P and EF-O, suggesting that enzymes involved in C and N cycling are more susceptible to particulate components than those involved in EF-P and EF-O.

The low response of EF-P to particulate components may have been due to the absence of roots in our mesocosms as rhizosphere bacteria and mycorrhiza are effectively solubilizing inorganic P (*Lladó, López-Mondéjar & Baldrian, 2017*). The observed changes in EF-C and EF-N with addition of particulate components may have been mainly due to changes in bacterial activity which has been shown to sensitively respond to particulate components (*Fierer et al., 2012*; *Freedman et al., 2013*; *Frey et al., 2014*). Importantly, the negative effects of N, Na and PAHs on EFs became neutral to positive with time of incubation, suggesting that soil microorganisms are resistant against particulate components and only respond if the input of particulate components continues for longer periods of time. However, the increased effects with time may also be related to increased biomass and diversity of bacteria at later stages of litter decomposition (*Voříšková & Baldrian, 2013*; *Purahong et al., 2014*; *Tláskal, Voříšková & Baldrian, 2016*).

**Earthworms stabilize soil enzymatic functions**

Both the beneficial as well as detrimental effects of particulate components on soil EFs diminished or even vanished in presence of earthworms, partly supporting our second hypothesis. The effect of *E. fetida* addition on soil moisture and microbial biomass was small in deciduous and coniferous forests (Fig. S12, S14). As epigeic earthworm species *E. fetida* does not form permanent burrows and therefore its effect on soil porosity is limited (*Brown, 1995*). However, the addition of *E. fetida* increased litter mass loss irrespective of the types of particulate components (*Yang et al., 2023*). Presumably, processing of litter by *E. fetida* facilitated microbial litter decomposition and mitigated detrimental effects of PAHs on soil EFs and microbial activity (*Klamerus-Iwan et al., 2015*). According to the resource allocation theory, the addition of nutrients may increase the secretion of extracellular enzymes by soil microorganisms in order to acquire nutrients from complex organic matter (*Sinsabaugh & Moorhead, 1994*; *Allison & Vitousek, 2005*). Earthworms likely increase microbial nutrient supply by secreting body mucus and accelerating litter decomposition (*Marhan & Scheu, 2006*; *Szlavecz et al., 2011*; *Hoang et al., 2016*), which may reduce the secretion of extracellular enzymes by microorganisms. The diminished response of microorganisms to the addition of N and Na in presence of earthworms therefore may reflect that they reduced the demand of microorganisms for nutrients.

In addition, with the addition of *E. fetida*, the effects of N and Na addition on soil pH were less strong in the deciduous than in the coniferous forest suggesting that earthworms stabilized soil EFs by mitigating effects of N and Na addition on soil pH. *Wang et al. (2021)* reported that up to 40% of the N deposited in forests may be lost *via* leaching within three months and this may be promoted by earthworms (*Frelich et al., 2006*). Although *E. fetida* has been shown to promote the binding of toxins such as PAHs to mineral surfaces (*Geissen et al., 2008*; *Rodriguez-Campos et al., 2014*), the addition of *E. fetida* did not mitigate the negative effect of PAHs on soil pH, but the negative effect of PAHs on soil microbial biomass was canceled out after 70 days. This suggests that during the first 70 days, *E. fetida* reduced the detrimental effect of PAHs *via* stimulating the binding of PAHs to mineral surfaces, but this effect diminished later, potentially due to reduced earthworm activity. Our correlation analyses further suggest that earthworms may stabilize soil EFs by affecting

multiple ecosystem functions related to litter decomposition, which is consistent with the findings of *Liu et al. (2019)*. Without earthworms, the addition of N, Na and PAHs enhanced the correlation between soil EFs of C and N and litter decomposition. The nylon bags we used to prevent colonization by earthworms may have reduced nutrient input from the litter layer into the microcosms. N and Na are utilized by microorganisms as nutrients, and certain microorganisms also use PAHs as carbon resources (*Wick, Colangelo & Harms, 2001*; *Wattiau et al., 2002*; *Pozdnyakova et al., 2018*), suggesting that in treatments without earthworms, particulate components may have stimulated the secretion of extracellular enzymes. In the N and Na addition treatments, earthworms weakened the correlation between soil EFs and litter decomposition. This suggests that increased nutrient supply due to earthworms (*Marhan & Scheu, 2006*; *Yang et al., 2023*) may have reduced the need for nutrients from decomposing litter by microorganisms and thereby reduced the secretion of extracellular enzymes and litter decomposition. By contrast, with the addition of PAHs earthworms strengthened the correlation between litter N loss and soil EF-N. Since PAHs may be utilized by soil microorganisms as carbon resource (*Wick, Colangelo & Harms, 2001*; *Wattiau et al., 2002*; *Pozdnyakova et al., 2018*), our results suggest that under PAHs pollution earthworms enhanced the acquisition of litter N by microorganisms through promoting the secretion of extracellular enzymes.

Although not replicated, our results suggest that the effects of earthworms on soil EFs differ between deciduous and coniferous forests. The addition of particulate components changed soil pH rather than soil microbial biomass and soil EFs in the coniferous forest. In the coniferous forest, the content of soil organic matter is low and litter decomposition is slower than in the deciduous forest (*Yang et al., 2023*). Further, correlations between soil EFs and litter decomposition were also less strong in the coniferous than in the deciduous forest. According to the resource allocation theory and stoichiometry, with the addition of inorganic N (or Na) carbon resources are needed to meet the demand of microorganisms for elements needed for enzyme secretion (*Sinsabaugh & Moorhead, 1994*; *Allison & Vitousek, 2005*). Considering the low soil carbon content in the coniferous forest, carbon resources may have limited the secretion of extracellular enzymes thereby preventing to respond to increased N and Na availability. Soil microorganisms of coniferous forests have been suggested to secrete more extracellular oxidases such as polyphenol oxidase due to high lignin content of coniferous litter (*Ji et al., 2020*). Thus, the less strong negative effect of PAHs on soil EFs in the coniferous forest may have been due to the ability of enzymes related to EF-O to decompose PAHs (*Hammel, Kalyanaraman & Kirk, 1986*). However, the less pronounced effect of earthworms on soil EFs in the coniferous forest also is related to the fact that particulate components little affected soil EFs in this forest. By contrast, the effect of earthworms on soil EFs in the deciduous forest was mainly in the treatments where particulate components modified EFs. Further, we added a lower number of earthworms to the coniferous than to the deciduous forest which also may have contributed to the less pronounced effect of earthworms in the coniferous compared to the deciduous forest. Previous studies suggested that the effect of earthworms on ecosystem functions weakens with lower earthworm abundance (*Cortez, 1998*; *Szlavecz et al., 2011*). Overall, our results

suggest that for stabilizing the functioning of microorganisms, a minimum number of earthworms might be needed.

## CONCLUSIONS

This study provided novel and detailed insight into how particulate components and soil fauna affect ecosystem functions in interactive ways. Our findings suggest that earthworms stabilize soil EFs irrespective of the type of particulate components, but that the mechanisms responsible for this stabilization vary among the types of particulate components. In addition to accelerating litter decomposition, earthworms also neutralized the effects of N and Na addition on soil EFs by affecting soil pH, but neutralized the effect of PAHs by affecting soil microbial biomass. The results further suggest that differences in soil microbial community composition and earthworm abundance are responsible for the differential effects of particulate components and earthworms on soil EFs in deciduous and coniferous forests. The results highlight the importance of studying the effects of soil fauna and particulate components on ecosystem functions in concert as earthworms may diminish or even cancel out the influence of atmospheric pollutants and stabilize ecosystem functions.

## ACKNOWLEDGEMENTS

We thank RM Ye, XY Zeng, Q Li, YL Ji, HJ Hu, TT Cao, H Jiang for field support.

### Funding

This work was supported by the National Natural Science Foundation of China (31870598, 32160356, 32001300); the Strategic Priority Research Program of the Chinese Academy of Sciences (A)(XDA19050400); the Jiangsu Forestry Science and Technology innovation and promotion project (LYKJ[2021]16); the Key Program of Scientific Research projects of Hunan Provincial Education Department (21A0334); the Key Specialized Research and Development breakthrough program in Henan Province (222102320289); the Scholarship of China Scholarship Council (202006190207), and the Open Access Publication funded by the University Göttingen. The funders had no role in study design, data collection and analysis, decision to publish, or preparation of the manuscript.

### Grant Disclosures

The following grant information was disclosed by the authors:
National Natural Science Foundation of China: 31870598, 32160356, 32001300.
Strategic Priority Research Program of the Chinese Academy of Sciences: (A)(XDA19050400).
Jiangsu Forestry Science and Technology innovation and promotion project: LYKJ[2021]16.
Key Program of Scientific Research projects of Hunan Provincial Education Department: 21A0334.

Key specialized research and development breakthrough program in Henan province: 222102320289.
Scholarship of China Scholarship Council: 202006190207.
University Göttingen.

## Competing Interests

The authors declare there are no competing interests.

## Author Contributions

- Junbo Yang conceived and designed the experiments, performed the experiments, analyzed the data, prepared figures and/or tables, authored or reviewed drafts of the article, and approved the final draft.
- Jingzhong Lu analyzed the data, prepared figures and/or tables, authored or reviewed drafts of the article, and approved the final draft.
- Yinghui Yang performed the experiments, authored or reviewed drafts of the article, and approved the final draft.
- Kai Tian conceived and designed the experiments, performed the experiments, analyzed the data, authored or reviewed drafts of the article, and approved the final draft.
- Xiangshi Kong performed the experiments, authored or reviewed drafts of the article, and approved the final draft.
- Xingjun Tian conceived and designed the experiments, authored or reviewed drafts of the article, and approved the final draft.
- Stefan Scheu analyzed the data, authored or reviewed drafts of the article, and approved the final draft.

## Data Availability

The raw data are available in the Supplementary Files.

## Supplemental Information

Supplemental information for this article can be found online at http://dx.doi.org/10.7717/peerj.15720#supplemental-information.

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
