# Peer review of "Earthworms neutralize the influence of components of particulate pollutants on soil extracellular enzymatic functions in subtropical forests"

_PeerJ, doi:10.7717/peerj.15720_

## Round 0.1 · original submission · Major Revisions

In this study, the effect of earthworm Eisenia fetida on the soil extracellular enzymatic functions in deciduous (Quercus variabilis) and coniferous (Pinus massoniana) forest that contaminated with ammonium nitrate, sodium chloride, and five mixed polycyclic aromatic hydrocarbons was investigated. The data has been presented clearly in both the manuscript and supporting files, but background information of the tested soils and discussion remains unclear which need further improvement.

1. The artificial run-off in Line 78 is not one of the ways for atmospheric deposition, and the N, Na, and PAHs were added with aqueous solutions in the field experiment which could not mimic the atmospheric deposition either. Therefore, the authors have to revise the related phases using more precise description about the effect of N, Na, and PAHs throughout the manuscript.

2. The authors need to clarify why different amounts of Eisenia fetida were added in the two tested forest field?

3. The reasons for the different performance between deciduous and coniferous forests are important for the application of this earthworm species. However, the authors only focused on soil pH, soil moisture, and the mass, C, and N loss of litter, yet did not provide sufficient discussion. More support from additional experimental results such as soil physiochemical properties and the characteristics of leaf litter would be useful for a comprehensive interpretation.

Reviewer 1 ·

Basic reporting

In this study, the authors conducted field experiment to test the influence of earthworms on the effects of deposited compounds on soil extracellular enzymatic functions in subtropical forests. It is interesting, but there are still some problems in my opinion:
1.Introduction is not clearly.
--It is lack of research background
--Lines 54-57, earthworms are considered as keystone species in soil as we've known. However E. fetida live in compost and is not a widespread species in soil as well, it can not be “a keystone species”.
--Lines 61-62, “……, e.g., the application of E. fetida to restore terrestrial ecosystems contaminated by deposited compounds.”, as I mentioned before, E.fetida is not a suitable earthworm species to restore terrestrial ecosystems actually. Why choose E.fetida in this study? Did the authors find E.fetida in soils at the end of experiment?
--What’s the problem about atomospheric depositions. Is atomospheric depositions really a serious problem in Forest? Xiao et al (2018) mentioned the effects of nutrient addition on enzyme activities, but actually they didn’t focus on “Atmospheric depositions”.
--Research progress need to be organized well again. Are there some previous studies about the relationship between earthworm and atomospheric deposition? The authors did not explain well the objective of the study. Why they conducted the field experiment in these two forests.
--Line 66-67 correct the format of these two references, please.
--Lines 92-94, add some new relative literatures here.
2.Materials and methods
--Line 117--add contents of organic matter and nutrients,please. It is better to show soil basic characteristics in this study. It’s very important to know the contents of N, Na, PAHS in soil as well.
--line 138, please replace “In the PAH treatment” with “In the PAHs treatment”. In line 291, please confirm that the “e.g.” in “……, and benefit e.g., from increased input of N ……” is incorrectly entered.
--line 143, “……, and the ratio of the five PAHs followed their ratio in the field”. it is really difficult to understand. What’s the real ratio in the field?
--Line 154-155, why did they inoculate different earthworm amounts in these two fields?
--Line171-180, how to prepare soil and litter for analyzing? put more details about soil enzyme analysis, please
3.Resutls:
-- pls show the abundance and biomass of E.fetida in different treatments, not only the ANOVA results. It is better to add the survival rate of earthworms in this study.
4.Discussion
--The discussion is very weak, the large number of references does not provide a clear interpretation of the data obtained.
--Line 297-307, this part is not very clear. I think the different effects on enzyme activities between different deposited compounds may not only due to their chemical properties but also soil physo-chemical and biological properties. It is better to analyze more indicators to explain it. “the plots closed by fine mesh bags may reduce the nutrient exchange” does not make sense. As you mentioned in Line 298, N and Na are both well soluble in water, so it is easy to transform from the mesh bags.
--Line 326-363 these parts are too long and not clear. It is better to use their nutrient data from this study to explain why earthworms stabilize soil enzymatic functions.
In summary, in my opinion, the present version of the manuscript should be major revised again.

Experimental design

no comment

Validity of the findings

no comments

Additional comments

no comments

Annotated reviews are not available for download in order to protect the identity of reviewers who chose to remain anonymous.

Reviewer 2 ·

Basic reporting

• Authors have nicely compiled the manuscript; both the abstract and introductions are sufficient and informative.

Experimental design

Although, methodology section informative and nicely written but few things need to be more clearer:
• In the experimental design section, 8 treatment combinations have been documented with two forest type and four deposits (2x4x4 = 8) replicated four times. But in the experiment earthworms (with and without) under both the forest types and deposits may increase the treatment combination to 16 (2x2x4=16) replicated four times. Same is visible from table ST-2 as well.
• Information about the concentration of N, Na and PAH is given but there is no such specific ration of dissolved five PAHs (used in present study) missing.

Validity of the findings

• Results of present investigation are representation nicely with tables, figures and statistical comparison. Findings are sufficiently supported under the discussion section with reasoning.

---

## Round 0.2 · Minor Revisions

The third reviewer has raised some minor comments on the Introduction and Discussion, as well as the presentation of data. Please address them carefully and make sufficient revisions.

Reviewer 2 ·

Basic reporting

Authors have incorporated suggestions.

Experimental design

Authors have incorporated suggestions.

Validity of the findings

Authors have incorporated suggestions.

Reviewer 3 ·

Basic reporting

This field experiment tested the interactive effects of an earthworm presence and the addition of different nutrients and pollutants on soil enzyme activities in two forests. The introduction was interesting and clear regarding the context of the study. The results presented are novel and important, as very few studies characterize how soil fauna modulate the impact of nutrients and pollutants in soils. The manuscript is generally well written, but I had issues with some of the interpretations stated in the discussion. In general, although the figures and tables are clear and well presented, the results are a bit difficult to follow, because there are a lot of treatments, enzymes, and temporal trends. I suggest to present an analysis at a coarser scale before digging into the detailed temporal trends: for example by averaging enzyme activities across different times to give an overall picture of the earthworms and treatments effects across time in the two forests. Given the main hypothesis that earthworm neutralize pollutant and nutrient effects, I also suggest to present with more details the results of the anova and significance of the interaction before detailing the different enzymes and temporal patterns. Please find below my comments to improve the manuscript.

Introduction

Minor comments
L105 the example given here states that pollutants can have positive effect on enzyme activities, while the start of the sentence argues that low concentrations can have no effect. I suggest to replace either the example showing non-significant effect of low concentration of Na, or changing the start of the sentence by stating that pollutants can have positive or negative effects depending on pollutants concentrations.

133 suggest change “how earthworms mediate the effects of pollutants on soil enzyme activities’
133. I am not sure if earthworms mediate pollutant effects, or rather can change the effect of pollutants on enzyme activities. Those are separate concepts. If there is a mediator role, than I suggest to introduce the mechanisms hypothesized for such mediation further here.

136 “because of the absence of natural ecosystem process and components “ is not fully clear to me. Please clarify here if the limitations are due to simplified laboratory experimental conditions.

144. In general I suggest to reinforce the introduction of mechanisms driving potential interactions between earthworms and pollutants here. Are earthworms affected themselves by pollutants? If the underlying hypothesis is that earthworms attenuate pollutants effects on microorganisms such as stated L151, is it possible to add a couple of sentences in the introduction about this?

171 should be “1 control, 3 particulate components x 2 earthworm treatments...”, or “4 particular components treatments including the control”

190 PAH concentrations are given as soil concentrations, but the treatments were aqueous solutions. I wonder if the soil concentrations were measured during the experiment, and how the PAH concentrations in the solution was established in order to reach the target PAH concentrations in the soil.

198 I don’t understand this statement “The N treatment doubled the rates of natural deposited compounds of N in the study region”, as L183 states that the N addition treatment was supposed to mimick “amounts of N deposited in the urban region of Nanjing”.

204 I suggest to homogenize the presentation of the compounds concentrations : there are several sentences about this that are currently scattered at different places in the paragraph which makes it difficult to follow. I suggest to make two separate paragraphs describing respectively the amounts of different compounds chosen and why, and a second paragraph about how the treatments were applied to the soil.

Given the low solubility of PAH I wonder if the treatments with a spray bottle really ensured that PAH reached the soil.

205 The earthworm treatment description given here does not match the brief description given above (L170) : absence/presence of earthworms is actually a low density vs high density treatment.

208. Please clarify here if the earthworm treatments with 60 and 20 individuals are for the deciduous versus coniferous forest, or if treatments with 60 vs 20 individuals were applied in each forest.

209. I don’t understand if the earthworms densities and biomass given here reflect the natural conditions in the forests or a result from the earthworm treatment that was applied. Please indicate the sample size associated with those means and SD values.

234. The main text should present a brief description of the methods used to measure soil enzyme activities, especially because this is at the core of the results presented here.

General comment on matmet: the temporal dimension is not super clear to me in this description, I suggest to add a statement at the start of the method section (e.g. L170 where the field experiment is described in general) specifying when the experiment was undertaken, and for how long.

Table 2. Suggest to add the time of earthworm sampling for these values (at the end of the experiment? Start?)

301 The results presented in Table 3 do not allow to compare the magnitude of particulate compounds effects in the two forests. I would rephrase the sentence to state that pollutant effects were only significant in the deciduous forest, not the coniferous one.

302 These increases are for the deciduous or coniferous forest?

303. in this paragraph, before presenting how earthworm effects differ between forest, I suggest to state clearly how earthworm affected soil enzymes (as this is the main question at hand): where there any significant interaction between polluants and earthworms? In which direction? For which enzyme types? Etc.

312 “PCA also indicated that earthworms reduced or neutralized the effects of …” I would like to see the resutls from the Anova for the interactive effect between earthworms and pollutants that is the best method to test such neutralizing effects statistically.

315 this sentence is redundant with sentence L303.

345. I am not sure to fully follow the correlation results and how they relate to the research questions. I suggest to add statements in the method section to justify why earthworms or pollutant treatments would affect the relationships between different soil properties.


Discussion

L366-371: the discussion starts with the study limitations. I would start by summarizing the main results and implications before presenting the caveats.

371; this statement only applies to the deciduous forest in the absence of earthworm though.

371-378. I would place this paragraph into the result section rather than the discussion : it summarizes well the main findings and effects found, and is not a discussion of findings in terms of their implications and relation with other literature.

381. this statement is not entirely accurate as Figure 1 shows that PAH decrease soil enzymes only on the short term (70days) in both control and earthworm treatments and only in the deciduous forest. Similarly, only the deciduous forest show increases in soil enzyme activities with N and Na treatments on the long term.

429. I disagree with this statement, because Figure 1 shows that PAH effects on enzymes decreased over time in the deciduous forest.

445. But the method section states that litter bags were placed on top of the mesocosms, I wonder if earthworm individuals were able to get out of the zip-locked mesocosms to process litter. Please clarify this in the method section.

447. I agree with the mechanism presented to explain the lower effect of PAH in the presence of earthworm, however, it is not clear to my why earthworms would neutralize the positive effect of nutrient addition. I would expect additive effects of nutrient addition and earthworm stimulation of microbial communities, or even a synergistic effect between them.

451. missing “on soil pH” at the end of the sentence

456. “This suggests that E. fetida in fact mitigated the detrimental effect of PAHs via stimulating the binding of PAHs to mineral surfaces.”
Given that the treatment was a repeated PAH application every month, doesn’t it means that this potential effect of E. fetida on binding of PAH to mineral surface increased over time?

458. I am not sure I completely follow this statement. As stated above, I found the correlation results not very clear. I suggest to elaborate a bit further on these results here, to clarify this statement.

488. This explanation does not fully convince me: If microorganisms were limited in resources in the coniferous forest, than I would expect that nutrient addition would have even stronger positive effects on enzyme activities than in the more fertile forest.

494-504. I am not sure that the earthworm abundance per se is responsible here, but the coniferous forest has a earthworm treatment of much lower intensity than the deciduous forest (the difference in earthworm abundance from 0 to 3 in the coniferous vs. 0 to 7 individuals/m2 in the deciduous).

Experimental design

no comment

Validity of the findings

no comment

Additional comments

no comment

---

## Round 0.3 · Minor Revisions

The authors have carefully responded to the comments and revised the manuscript accordingly in the 2nd revised version. Some of the responses need further improvement and verification before consideration of acceptance for publication.

1. L116. The authors replied that they avoided using the term ‘mediate’ but still used it here. Please remove this word and revise this sentence.

2. L160. As suggested by the reviewer, the statement should be ‘4 particular components treatments including the control x 2 earthworm treatments x 4 replicates’, because the control treatments didn’t have any additional particular components.

3. L167-168. Why did the installment of the mesocosms take place for two-three months? Please clarify the purposes.

4. L179-180. As commended by the reviewer, it is not convincing to use the unit of ‘kg-1 dry soil y-1’ to present the concentrations of PAHs in this study, because the authors added PAHs with solution into the soil in field. Please rewrite this sentence.

5. L199-201. The authors haven’t replied to the reviewer’s comment appropriately. The description in this sentence is the results of the earthworm treatments after a 365-day experiment? Or investigation of the field before the experiment? Please clarify.

6. Table 2. The data for ‘Initial’ should be placed in the columns of ‘+Earthworms’ not in those of ‘-Earthworms’, right? The authors claimed that they removed all earthworm individuals during set up of the mesocosms, and they presented ‘0’ in the ‘Control’ treatments as well.

7. L202-205. As commended by the reviewer, ‘I wonder if earthworm individuals were able to get out of the zip-locked mesocosms to process litter’. The authors haven’t replied to this question. Please add more details.

8. L209. Why did it take three-four months to count the earthworms after the 365-day experiment? Please clarify.

9. L297-298. ‘Futher, in total N and Na ...’ This sentence doesn’t sound acceptable in grammar. Please rewrite it.

10. L458. What does ‘nutrient mining’ mean? Please revise this confusing phrase.

11. L463. The wording ‘by of’ doesn't make sense and should be corrected.

---

## Round 0.4 · accepted · Accept

The authors have revised the manuscript appropriately based on the comments from the reviewers and editor, and the current version of this manuscript is ready for publication.